# Comparison of Small Blood Vessel Diameter with Intravascular Ultrasound and Coronary Angiography for Guidance of Percutaneous Coronary Intervention

**DOI:** 10.3390/diagnostics14121312

**Published:** 2024-06-20

**Authors:** Sergio A. Zuñiga-Mendoza, Emanuel Zayas-Diaz, Victoria R. Armenta-Velazquez, Ana A. Silva-Baeza, Juan J. Beltran-Ochoa, Misael A. Medina-Servin, Maria G. Zavala-Cerna

**Affiliations:** 1Hospital Regional Valentin Gomez Farias, ISSSTE, Guadalajara 44340, Jalisco, Mexico; sergio.zuniga@edu.uag.mx (S.A.Z.-M.); dr_beltran8a@hotmail.com (J.J.B.-O.); misaelmedinamd@gmail.com (M.A.M.-S.); 2Unidad Académica Ciencias de la Salud, Universidad Autónoma de Guadalajara, Zapopan 45129, Jalisco, Mexico; emanuel.zayas@edu.uag.mx (E.Z.-D.); victoria.armenta@edu.uag.mx (V.R.A.-V.); anaa.silva@edu.uag.mx (A.A.S.-B.)

**Keywords:** cardiovascular disease, coronary artery disease, percutaneous coronary intervention, coronary angiography, intravascular ultrasound, small vessel coronary artery disease

## Abstract

Major cardiovascular events (MACEs) are a cause of major mortality worldwide. The narrowing and blockage of coronary arteries with atherosclerotic plaques are diagnosed and treated with percutaneous coronary intervention (PCI). During this procedure, coronary angiography (CAG) remains the most widely used guidance modality for the evaluation of the affected blood vessel. The measurement of the blood vessel diameter is an important factor to consider in order to decide if stent colocation is suitable for the intervention. In this regard, a small blood vessel (<2.75 mm) is majorly left without stent colocation; however, small vessel coronary artery disease (SvCAD) is a significant risk factor for the recurrence of MACEs, maybe due to the lack of a standardized treatment related to the diameter of the affected blood vessel; therefore, a more precise measurement is needed. The use of CAG for the measurement of the blood vessel diameter has some important limitations that can be improved with the use of newer techniques such as intravascular ultrasound (IVUS), although at higher costs, which might explain its underuse. To address differences in blood vessel diameter measurements and identify specific cases where IVUS might be of additional benefit for the patient, we conducted a retrospective study in patients who underwent PCI for MACEs with affection for at least one small blood vessel. We compared the measurements of the affected small blood vessels’ diameter obtained by CAG and IVUS to identify cases of reclassification of the affected blood vessel; additionally, we underwent a multivariate analysis to identify risk factors associated with blood vessel reclassification. We included information from 48 patients with a mean ± SD age of 69.1 ± 11.9 years; 70.8% were men and 29.2% were women. The mean diameter with CAG and IVUS was 2.1 mm (95% CI 1.9–2.2), and 2.8 (2.8–3.0), respectively. The estimated difference was of 0.8 mm (95% CI 0.7–0.9). We found a significant positive low correlation in diameter measurements of small blood vessels obtained with CAG and IVUS (r = 0.1242 *p* = 0.014). In total, 37 (77%) patients had a reclassification of the affected blood vessel with IVUS. In 21 cases, the affected blood vessel changed from a small to a medium size (2.75–3.00 mm), and in 15 cases, the affected vessel changed from a small to a large size (<3.00 mm). The Bland–Altman plot was used to evaluate agreement in measurements with CAG and IVUS. The change in blood vessel classification with IVUs was important for the decision of intervention and stent collocation. The only variable associated with reclassification of blood vessels after adjustment in a multivariate analysis was T2D (type 2 diabetes) (*p* = 0 0.035). Our findings corroborate that blood vessels might appear smaller with CAG, especially in patients with T2D; therefore, at least in these cases, the use of IVUS is recommended over CAG.

## 1. Introduction

Cardiovascular disease (CVD) is a cause of major mortality worldwide, characterized by a reduction in blood supply to the myocardium, commonly due to a narrowing or blockage of coronary arteries, causing a mismatch between myocardial oxygen supply and demand [1,2]. Chest pain, dyspnea, dizziness, and nausea are common symptoms that patients with a coronary artery disease (CAD) may present. Commonly, the narrowing or blockage of the coronary arteries is due to atherosclerosis, which is also responsible for CVD-related deaths [3]. Atherosclerotic coronary plaque destabilization and progression are responsible for both acute and chronic coronary syndromes, as well as sudden coronary death [4,5].

The narrowing and blockage of coronary arteries can be diagnosed and treated with percutaneous coronary intervention (PCI) [6,7]. The goal of PCI is to improve the minimum lumen diameter in a given coronary segment, which is defined as the average of the diameters of apparently normal segments localized proximally and distally to the target segment [8]. To achieve revascularization, and irrespective of the available approach, the reference diameter is an important parameter, required for intervention during percutaneous treatment, which is obtained from an average of proximal and distal diameters [9]. A small reference diameter may indicate a true small coronary artery, a large plaque burden, or the presence of diffuse disease [10].

Small vessel coronary artery disease (SvCAD) is a significant risk factor for major adverse cardiovascular events (MACEs) in PCI. Currently, there are discrepancies related to the definition of a small vessel. Several thresholds of maximum lumen size have been described that range from <3.0 to <2.25 mm, and this lack of consensus causes heterogeneity both in the results of clinical studies and, consequently, in discrepancies related to the appropriate therapeutic approach [11]. SvCAD is often diffuse, and revascularization should only be performed in patients with confirmed ischemia and in hemodynamically significant lesions based on functional assessment due to the risks associated with the manipulation of small blood vessels [12]. Additionally, the instrument selected for the revascularization needs to be adapted to the size of the blood vessel.

Among the options for revascularization are contemporary second-generation drug-eluting stents (DESs). Their use has been associated with superior performance in patients with small vessel CAD, resulting in a reduction in both lumen loss and clinical restenosis [13]. Similarly, drug-coated balloons (DCBs) provide a fast and high-dose delivery of antiproliferative drugs to the vessel wall, which carries several anticipated benefits over DESs, such as the lack of a permanent scaffold and the need for only a short-term prescription of dual antiplatelet therapy [13].

During PCI, coronary angiography (CAG) remains the most widely used guidance modality for making decisions about the intervention of damaged blood vessels [2]. However, there are some limitations, including the projection of two dimensions for lumen artery determination, as well as its limited ability to assess blood vessel wall, poor definition of plaque extension, and limitations in the ability to define the characteristics and degree of atherosclerosis [14,15]. Moreover, in the presence of diffuse coronary artery disease, the reference blood vessel diameter may not be easily discernable, and the true vessel diameter can easily be underestimated with the use of CAG [16].

To overcome these limitations, modern technologies have been developed, such as intravascular ultrasound (IVUS) [17]. The large-scale prospective study for the assessment of dual antiplatelet therapy with drug-eluting stents (ADAPT-DES) demonstrated that IVUS compared to angiography guidance was associated with a reduced 2-year rate of MACEs, including restenosis and myocardial infarction [18]. Furthermore, a recent meta-analysis reported that in comparison with CAG, IVUS significantly reduced the development of MACEs as well as target vessel/lesion revascularization (TVR/TLR) [17].

Despite IVUS’s superiority, its use for guidance remains low, as an estimate from the US (United States) Medicare cohort between 2009 and 2017 described its use in up to 5.6% of PCI procedures [19], which might be the result of low reimbursement for IVUS use compared to CAG for PCI guidance. However, the improvement in clinical outcomes and the prevention of complications should be emphasized.

The primary aim of this study was to investigate the change in diameter measurement of IVUS-guided PCI as compared to angiographic measurements in patients with SvCAD. A secondary aim was to evaluate the presence of associations between clinical comorbidities with major variation in the estimation of the small blood vessel diameter.

## 2. Materials and Methods

This was a retrospective study that included patients who underwent PCI for CAD from January 2021 to December 2022.

### 2.1. Study Population

Registries of patients were included from those who attended the hemodynamic clinic in the tertiary hospital “Hospital Regional Dr Valentín Gómez Farías del Instituto de Seguridad y Servicios Sociales para los Trabajadores del Estado (ISSSTE)” in Zapopan, Jalisco. To be included in this study, patients had to fulfill the following inclusion criteria: age > 18; any gender; and having a CAD with the affection of at least one small blood vessel, defined as a reference vessel diameter of <2.75 mm by angiographic PCI guidance.

Patients with incomplete medical chart information, severe calcified and/or ostial lesions, and patients that required revascularization surgery, valvular exchange, or aortic surgery were excluded from this study. The institutional review board revised and approved the protocol. Exemption from informed consent was granted given the retrospective design of this study.

### 2.2. Study Outcomes

Coronary angiography (CAG) was performed using a standard percutaneous approach through the femoral artery, unless this artery was unavailable, following the intracoronary administration of 100 to 200 μg of nitroglycerin (NTG). A reference vessel diameter (RVD) of <2.75 mm in the targeted lesion was defined as a small coronary vessel, as stated by a previously established definition of a small blood vessel [8,20].

The minimum lumen diameter (MLD) and lesion length were also recorded. The diameter percentage of stenosis was calculated as the ratio between the MLD and the RVD.

Intravascular ultrasound (IVUS) was performed immediately after the CAG in the targeted small blood vessel with a 40 MHz IVUS coronary imaging catheter. The catheter was advanced distally into the target vessel as far as possible and then automatically pulled back; image acquisition and posterior analysis were performed using computerized planimetry for every 1 mm of axial length. Lumen and external elastic membrane cross-sectional areas were measured. The lesion site was the image slice with the smallest cross-sectional lumen area. The proximal and distal reference segments were the most normal-looking segments within 5 mm proximal and distal to the lesion, following the criteria of the American College of Cardiology Clinical Expert Consensus Document on IVUS [10].

CAG and IVUS analyses were performed by two independent observers, and decisions for the selection of the stent or balloon were made in consensus. A third physician was used in the case of differing opinions on treatment.

Following PCI, all patients were treated with clopidogrel 75 mg/day for 12 months and aspirin 100 mg/day permanently.

Other medications were prescribed or adjusted according to patients’ previous conditions and comorbidities.

### 2.3. Statistical Analysis

Categorical variables were presented as frequencies and percentages; continuous variables were expressed as means ± standard deviations (SDs). Differences in the diameter of vessels and minimal lumen diameter were analyzed by the Mann–Whitney U test. Correlations between measurements by angiography and IVUS were analyzed by Spearman test. Then, patients were grouped according to their need for blood vessel reclassification and analyzed with the Chi square test; finally, to address risk factors for blood vessel reclassification, we performed a multivariable logistic regression. The Bland–Altman plot was used to illustrate the agreement between diameter measurements obtained with CAG and IVUS, in which the difference between the minimal lumen diameter was plotted against the mean value [21]. A probability value < 0.05 was considered significant. STATA 17 was used for calculations.

## 3. Results

Between January and September of the year 2022, a total of 48 patients with CAD with the affection of at least one small coronary blood vessel (<2.75 mm) were included in our study. The mean ± SD age was 69.1 ± 11.9 years; 34 (70.8%) were men and 14 (29.2%) were women. Clinical data from patients were analyzed, and multiple comorbidities were reported; 6.3% had previous acute myocardial infarctions (AMIs). This information is displayed in Table 1.

The minimal lumen diameter of affected small blood vessels was measured with CAG and IVU (intravascular ultrasound) (Figure 1).

We found a small but significant correlation (r = 0.1242; *p* = 0.014) depicted in Figure 2. Importantly, the correlation is more significant in blood vessels that have a minimal lumen diameter > 2 mm (about 0.08 in) (Table 2).

The reported mean and 95% CI for the minimal lumen diameter with CAG was 2.1 (1.9–2.2), and for IVUS, it was 2.8 (2.8–3.0), obtaining a difference of 0.8 (0.7–0.9) *p* < 0.0001, with an infra estimation of the real minimal lumen diameter by CAG. The agreement between blood vessel reference diameter measured by CAG compared to IVUS in the whole group with a fixed bias of −0.2556 mm (95%CI −0.6618 to −0.1505; *p* = 0.212; LOA = −0.1876 to −0.2556; Figure 3).

Out of the total sample, 37 patients’ affected blood vessels (77%) were reclassified. A total of 21 patients’ (44%) affected blood vessels were reclassified as median-sized (2.75–3.00 mm), and 15 patients’ (31%) affected blood vessels were reclassified as great-caliber blood vessels (3.00 mm). In particular, this last group had the biggest change in terms of the treatment, either by the decision to use a different instrument or by the selection of the proper stent.

After the PCIs, there was no coronary perforation, serious aortic dissection, or acute target blood vessel occlusion in any of the analyzed cases. A stent was collocated in 46 cases, and the use of a medicated balloon was carried out in two cases. The final selection of stent diameter and high-pressure balloons used during post-dilation or medicated balloons was made according to the vessel diameter obtained by IVUS. The stent (mean ± SD) diameter was 2.7 ± 0.5. Table 3 details the characteristics of treatment for the population.

After the observation of a minimal correlation, blood vessels were reclassified according to their IVUS minimal lumen diameter. Afterwards, we searched for variables that could be associated with a change in the classification of the affected vessel based on the measurement of the minimal lumen diameter by IVUS. Table 2 represents the frequency of variables previously attributed to restenosis or small vessel coronary artery disease. Variables were analyzed separately, and variables with a significant difference in the distribution associated with a reclassification in the affected blood vessel are presented.

We then performed a multivariate analysis to test a model with a combination of variables to better explain the reclassification or if the strength of association with T2D (type 2 diabetes) could change. Table 3 depicts the results from this multivariate analysis (*p* = 0.1506).

## 4. Discussion

We found a low but significant correlation in the reference blood vessel diameter, measured by CAG and IVUS. A difference of 0.8 mm was found in our study; this difference has been confirmed by others [22,23]. However, these studies only included patients with T2D, and other risk factors were not evaluated. The reasons for the discrepancy in measurements related to each technique can be explained by limitations previously described in association with CAG or the presence of a diffuse coronary artery disease, with systemic endothelium dysfunction associated with T2D, hypertension, and other conditions, which tends to imply difficulties in the true measurement diameter. Additionally, the potential distal location of the small vessel, proximal tortuosity, and location at bifurcation points can also contribute to the differences registered with the use of each technique.

Previous studies in the small arteries and arterioles of diabetic subjects have demonstrated that morphological changes are preceded by vasomotor dysfunction, which is the result of the affection of both smooth muscle- and endothelium-mediated regulatory mechanisms that include an abnormal level of nitric oxide (NO) production [24].

When analyzing the endothelium damage related to diabetes, it has been described that hyperglycemia suppresses flow-mediated endothelial-dependent vasodilation, which has been demonstrated in both patients with diabetes and healthy subjects with induced hyperglycemia [25].

Furthermore, chronic hyperglycemia induces an inflammatory cascade, mainly characterized by the activation of several pathways, including diacylglycerol (DAG) and protein kinase C (PCK), which later can be associated with an increase in the oxidative stress associated with endothelial dysfunction [26] and cumulative long-term changes in the structure and function of macromolecules through the formation of advanced glycation end products (AGEs) [27].

Moreover, endothelial cells express insulin receptors, which can trigger NO-dependent vasodilation by increasing its production as well as the production of endothelium-derived relaxing factors (e.g., PGI2) [28].

Endothelial dysfunction in T2D patients seems to be multifactorial. Additional molecules described in association with this phenomenon include the inflammatory cytokines tumor necrosis factor-alpha (TNF-a) and IL-6 [29,30], the peroxisome proliferator-activated receptor-g (PPAR-g) [31], an increased generation of oxygen-free radicals by NAD(P)H-dependent oxidases, xanthine oxidase, lipoxygenase, mitochondrial oxidase, and NOS, as well as the diacylglycerol/PKC (DAG/PKC) pathway [32].

Furthermore, T2D has been associated with decreased stiffness of coronary resistance microvessels (CRMs), as a result of reduced stiffness in vascular smooth muscle cells (VSMC), determined by a computational model, with the analysis of male T2DM homozygous db/db mice and heterozygous non-diabetic control Db/db mice, where vascular smooth muscle cells (VSMCs) and coronary resistance microvessels (CRMs) were collected, cultured, and studied by atomic force microscopy and state-of-the-art traction force microscopy, which showed that diabetic coronary VSMCs had reduced stiffness, decreased adhesion to fibronectin, and increased tensile force properties, which, together, are mechanistically revealing the underlying causes of the altered mechanical and contractile properties of intact diabetic CRMs [33].

The mechanisms involved in the etiology of microvascular complications associated with T2D have been described in isolation, and it rather seems that a combination of several of the previous mechanisms described is responsible for the endothelial dysfunction; the common final pathway seems to be an increase in oxidative stress with impaired vasomotion.

Another important previously attributable risk factor for small blood vessel reclassification is active smoking, since 0.3–5.3% of cases have been associated with coronary ectasia, which consists of a pathological remodeling characterized by a diffuse dilation of the coronary artery diameter of >1.5 times [34]. However, in our study, active smoking was not associated with vascular reclassification.

Hypertension has been associated with vascular reclassification as well, majorly due to a phenomenon termed early vascular aging (EVA), which has been described as an arterial stiffening in the middle layer of the large elastic arteries, a process that can be measured by pulse wave velocity [35]. Factors that can trigger such stiffness include hypertension per se but also chronic inflammatory conditions such as rheumatoid arthritis or inflammatory bowel disease. Although hypertension would be mostly associated with rigidity rather than endothelium elasticity, in our study, hypertension was not associated with an increased risk of reclassification; however, 64.6% of our studied population had hypertension, and the reason why we did not find an association could be due to the high numbers of patients with hypertension in both groups. A study with more patients could clarify a real association or an additive behavior along with T2D.

The exact dimension of the lumen diameter of the affected vessel is a key factor in determining the size of the stent and balloons utilized during the procedure; this ensures appropriate treatment of small-vessel coronary lesions as well as a better evaluation of the atherosclerotic plaque. It is for this main reason that IVU guidance has been associated with positive clinical outcomes, with a declined incidence in major adverse cardiac events (MACEs) and in-stent restenosis (ISR) in T2D patients with CAD [23].

The superiority of IVUS compared to angiography has been well documented in the past after the conduction of both observational and randomized controlled trials, where IVUS has been associated with better clinical outcomes, including a decreased incidence of target vessel failure after 12 months and mitigation of target lesion revascularization and stent thrombosis [36,37]. Furthermore, recent evidence derived from a meta-analysis comparing the outcomes of patients with CAD after the use of IVUS versus CAG for PCI demonstrated that the use of IVUS reduced significantly the risk for MACEs from 13.3% to 8% with a RR of 0.63 and 95% CI of 0.54–0.73, cardiac death (RR 0.47; 95% CI 0.31–0.73), definite probable stent thrombosis (RR 0.48; 95% CI 0.24–0.97), targeted vessel revascularization (RR 0.62; 95% CI 0.46–0.83), and target lesion revascularization (RR 0.61; 95% CI 0.47–0.79). However, no difference was found in all-cause death (RR 0.79; 95% CI 0.53–1.18) and myocardial infarction (RR 0.80; 95% CI 0.61–1.04) [38].

Despite the clinical benefits associated with the use of intravascular ultrasound (IVUS) guidance for PCI [9], most patients with coronary artery disease still undergo guidance by percutaneous angiography in the real-world setting, majorly due to a low reimbursement of IVUS use compared to CAG [19]. Given this reluctance in the guidance for PCI by IVUS, an important consideration should be made for patients with diabetes, irrespective of the type of plaque or the presence of additional comorbidities or clinical characteristics.

Importantly, we recognized that when angiography is used alone for the guidance of PCI, the residual clearance between the mesh stent and the intima after the plaque rupture results in the expansion of the balloon, and this space can be filled with the contrast agent, resulting in a false impression of completion, which is avoided with IVUs [39]. Moreover, a coverage failure of the full segment-affected length with the stent has been associated with an increased risk of subacute stent thrombosis and late restenosis [40].

The association between T2D and coronary artery disease is non-debatable and has been widely demonstrated over the last two decades, especially in population-based studies, where the 7-year incidence of first myocardial infarction or death was 20% for diabetic patients but only 3.5% for non-diabetic patients [41]. However, this association still needs clarification on the physio-pathological pathways and, most importantly, those associated with therapeutic decisions in the CAD scenario.

Our study presents several limitations; some of them were related to the design since this was an observational study, there was no randomization, and selection bias could be present. Another limitation is the number of patients included in the present study, which was limited to 48 patients, and this could represent a lack of power to analyze the associations of previous attributable risk factors for SvCAD, with the probability of blood vessel reclassification. Larger studies with experimental designs could help in the confirmation of our results and a more powerful analysis of risk factors.

## 5. Conclusions

PCI treatment for small coronary vessels remains debatable based on the absence of an agreement to define a true small blood vessel and the fact that small blood coronary vessels are often considered not treatable with a stent because of the lack of clinically relevant outcomes, the high restenosis rate, and difficulties associated with implementing a stent in a small vessel. We confirmed that PCIs for patients with affection of a small blood vessel (<2.75 mm) should be guided by IVUs, mostly in the presence of diabetes, since other previously attributed risk factors for PCI were not significantly associated with the reclassification of blood vessels.

## Figures and Tables

**Figure 1 diagnostics-14-01312-f001:**
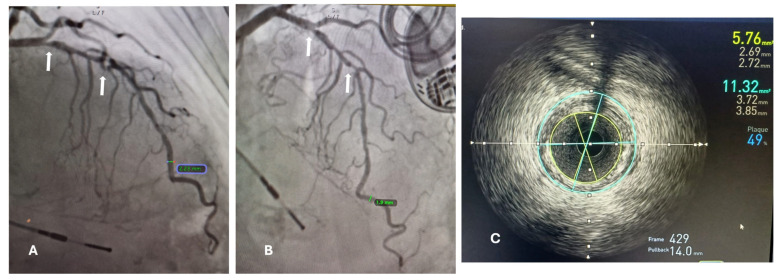
(**A**) Coronary angiography findings show 85% stenosis in the mid-left anterior descending coronary artery (white arrows). The diameter of the affected vessel was measured in the most normal-looking segments distal from the lesion, reported as 2.23 mm. (**B**) Coronary angiography with a lateral view of the same vessel, where the most distal lesion is better appreciated; the vessel diameter in this view (distal segment) was 1.9 mm. (**C**) IVUS-guided diameter measurement of the same vessel: the blue circle represents measurement of the reference vessel diameter (RVD) of 3.85–3.72 mm, and the yellow circle represents the remaining luminal area as a result of an obstruction of 49% due to an atherosclerotic plaque, which is the minimal lumen diameter (MLD) of 2.69–2.72 mm.

**Figure 2 diagnostics-14-01312-f002:**
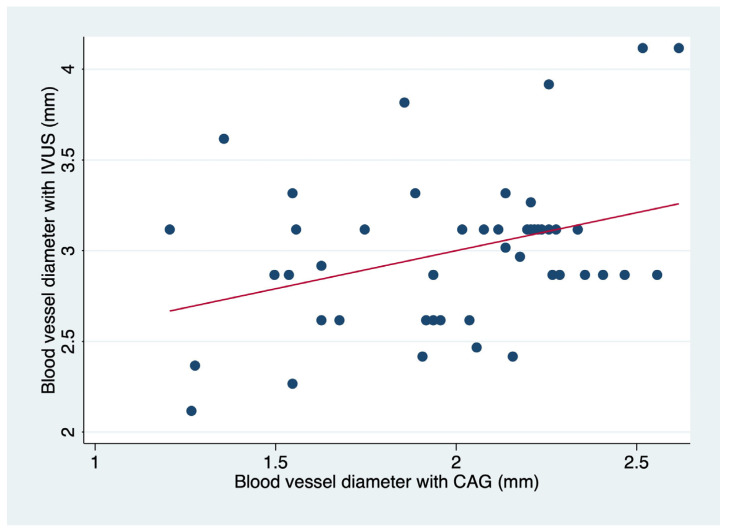
Spearman correlation for minimal lumen diameter in affected blood vessels by CAG (coronary angiography) and IVUS (intravascular ultrasound).

**Figure 3 diagnostics-14-01312-f003:**
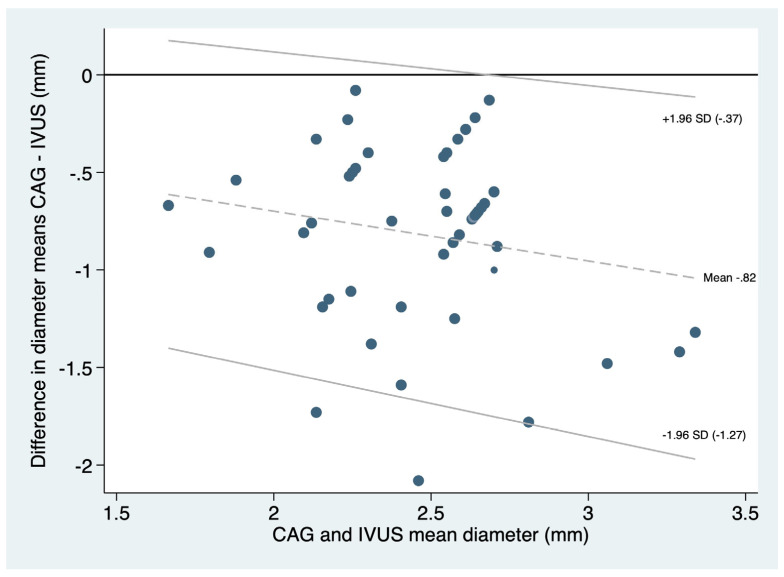
Bland–Altman plot demonstrating the degree of agreement between CAG and IVUS in measuring the small blood vessel reference diameter.

**Table 1 diagnostics-14-01312-t001:** Demographic and clinical characteristics of patients with CAD.

Characteristic	n = 48
Age	69.1 ± 11.9
Men	34 (70.8)
Women	14 (29.2)
T2D	21 (43.8)
Hypertension	31 (64.6)
Dyslipidemia	3 (6.3)
Chronic kidney disease	2 (4.2)
Obesity (BMI ≥ 30)	5 (10.4)
Current smokers	11 (22.9)
Previous AMI	3 (6.3)
Previous PCI	8 (16.7)
Stable angina	20 (41.7)
Unstable angina	6 (12.5)
NSTEMI	6 (12.5)
STEMI	12 (25.0)
Affected blood vessel	
Anterior descendant	32 (66.7)
Right coronary	11 (22.9)
Circumflex artery	3 (6.3)
Intermediate branch	1 (2.1)
Posterior descendant	1 (2.1)

Information is presented as mean ± SD or n (%). Abbreviations: T2D: type 2 diabetes; BMI: body mass index; AMI: acute myocardial infarction; PCI: percutaneous coronary intervention; NSTEMI: myocardial infarction without ST segment elevation; and STEMI: myocardial infarction with ST segment elevation.

**Table 2 diagnostics-14-01312-t002:** Demographic and clinical characteristics associated with vascular reclassification.

Characteristic	No Change (n = 11)	Reclassified Blood Vessel (n = 37)	*p*-Value	Univariate OR (95% CI)
Age (mean ± SD)	68.0 ± 10.6	69.5 ± 12.4	0.714	1.01 (0.95–1.07)
Men (frequency %)	8 (73)	26 (70)	0.875	0.88 (0.20–3.98)
Obesity	1 (13)	4 (11)	0.870	1.21 (0.12–12.12)
T2D	4 (36)	13 (35)	0.036 *	0.20 (0.05–0.90)
Hypertension	8 (73)	23 (62)	0.522	0.62 (0.14–2.72)
Active smoking	1 (13)	10 (27)	0.239	3.70 (0.42–32.77)
Previous AMI	0	3 (8)	0.001 *	1 (1.57–6.10)
Fibro calcic plaque	7 (64)	25 (68)	0.808	1.19 (0.29–4.87)
Fibro lipidic plaque	3 (27)	9 (24)	0.843	0.86 (0.19–3.94)

T2D: type 2 diabetes; AMI: acute myocardial infarction. * Significant *p*-value.

**Table 3 diagnostics-14-01312-t003:** Multivariate analysis model for the reclassification of affected blood vessels in patients with CAD.

Variable	OR (CI 95%)	*p*-Value
Age	1.08 (0.97–1.20)	0.134
T2D	0.05 (0.01–0.81)	0.035 *
Hypertension	0.54 (0.05–5.88)	0.620
Obesity	0.08 (0.01–2.66)	0.159
Active smoking	6.46 (0.25–170.15)	0.264
Previous AMI	1.00	-

* Significant *p*-value.

## Data Availability

All relevant data are included in the manuscript.

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
