# Peer review of "Comparison of Small Blood Vessel Diameter with Intravascular Ultrasound and Coronary Angiography for Guidance of Percutaneous Coronary Intervention"

_diagnostics, 2024, doi:10.3390/diagnostics14121312_

Round 1
Reviewer 1 Report
Comments and Suggestions for Authors
This study addresses an interesting topic and highlights significant results. However, I have identified some notable shortcomings that need to be corrected to improve the quality of the manuscript:
1. In line 160 it is stated that 34% are of the masculine gender, while the table indicates that this corresponds to 34 individuals, which is 70.8%. It is necessary to reconcile these data
2. When the mean and standard deviation are reported, they are usually reported as mean±SD. Authors should correct this throughout the whole manuscript.
3. When comparing two methods as in this study, it is appropriate to use a Bland-Altman plot. The authors should consider presenting their data using this plot instead of a scatter plot.
4. Logistic regression is lacking some important information. What are the values of the univariate OR for the variables examined? Was this model statistically significant?
5. In Discusion section, limitations of the study should be stated
Author Response
- In line 160 it is stated that 34% are of the masculine gender, while the table indicates that this corresponds to 34 individuals, which is 70.8%. It is necessary to reconcile these data
Response: Thank you for addressing this mistake, the change was made.
- When the mean and standard deviation are reported, they are usually reported as mean±SD. Authors should correct this throughout the whole manuscript.
Response: We agree with this, in the original version of the manuscript we used “+” but I guess the underlined got deleted at some point during the submission, we have used the symbol ± instead of + to avoid these changes when formatting.
- When comparing two methods as in this study, it is appropriate to use a Bland-Altman plot. The authors should consider presenting their data using this plot instead of a scatter plot.
Response: Thank you for this suggestion. We tried to generate a Bland-Altman plot using STATA, however an error code stated the following: could not calculate numerical derivatives due to discontinuous region with missing values encounteredr(430). I think the data can´t be presented in this graph as it is, it will need to be “adjusted”, honestly not sure if this is the best way to present the information, if the reviewer could add more information on how to create this graph, we would be happy to explore the best way to analyze and present the information.
Our data is available upon reviewers’ request.
- Logistic regression is lacking some important information. What are the values of the univariate OR for the variables examined? Was this model statistically significant?
Response: Thank you for this comment. We added a column to table 2 with univariate OR (95% CI), some adjustments were made to this table, as we did again the tests and identify some previous mistakes. The model was not statistically significant, and this information was added to the text in line 460.
- In Discusion section, limitations of the study should be stated
Response: Thank you for this observation. We have added limitations of our study to lines 628-634.
Reviewer 2 Report
Comments and Suggestions for Authors
Thank you for the opportunity to review this paper. Understanding the methods of visualizing the dimensions of coronary arteries can help treatment interventions. This study examined intravascular ultrasound and angiography as well as linkages between co-morbidities and small blood vessel diameter.
Abstract- Perhaps some English language edits to help reader comprehension is warranted.
- Line 31 .. significant positive low correlation between..... ??? identify variables.
- DT2 is undefined for the reader
Introduction
There could be more information regarding the validity of IVUS as a tool. As I read the intro and subsequent results, I am not convinced that one must accept IVUS as the standard. This is probably due to the lack on information in the introduction to show the validity of IVUS compared to angiography. Perhaps more information from ref 20,21?
Methods
Can you provide rational for the small vessel classification of < 2.75mm? Was this based on any particular standard, or averaging of ranges... Please explain.
Is there a reason to specify the proportion of masculine gender? This was also evident in the analysis. Why not simply identify the proportion of men and women? Unclear to me why masculine gender is specified only.
Results
Table 2. Perhaps as a bottom footnote on the table you can identify the items that are abbreviated. DT2, AMI STU, AMI STE.... for the reader it is best to identify all abbreviations.
figure 1 line 172- Typo (I make this one all of the time).. "form" vs from
Figure 2 and subsequent discussion
r=.12 is a very low correlation despite statistical significance. This makes me wonder if PCI and IVUS are actually measuring similar things, as there is virtually no relationship between the measurements. Could this be the methods of taking the measures?
I think this is a section to provide more discussion. You seem to focus on the statistical relevance (<.05) rather than the very poor correlation of two methods that are supposedly measuring the same thing. Would one expect such a low correlation?
typo- the last two lines of the manuscript (lines 318,319) I believe are not meant to be there.
Comments on the Quality of English LanguageCheck for minor grammar edits
Author Response
Abstract- Perhaps some English language edits to help reader comprehension is warranted.
Response: Thank you, the abstract was reviewed and improved. We hope that with the changes comprehension of our work is warranted.
- Line 31 .. significant positive low correlation between..... ??? identify variables.
Response: Line 31 now is line 37, we added information about the variables analyzed in the correlation.
- DT2 is undefined for the reader
Response: we have corrected this abbreviation to T2D and define what it stands for on line 44.
Introduction
There could be more information regarding the validity of IVUS as a tool. As I read the intro and subsequent results, I am not convinced that one must accept IVUS as the standard. This is probably due to the lack on information in the introduction to show the validity of IVUS compared to angiography. Perhaps more information from ref 20,21?
Response: Thank you for your observation, we agree that there is still lack of conclusive evidence, and this was exactly what drove our hypothesis: A more specific measurement could be obtained with IVUS, however the difference was not clearly established in our population with our equipment’s, and neither if there could be clinical characteristics associated with a higher difference in the measurements. After your observation, we underwent a search for more information, and found a couple of papers that talked about outcomes when comparing CAG to IVUS, we’ve added this information to the discussion section (lines 488-543). We also change the order in which the information was presented in the introduction section, a more clear flow of the information is presented with this changes, and the addition of more information, hopefully the idea is more clear.
Methods
Can you provide rational for the small vessel classification of < 2.75mm? Was this based on any particular standard, or averaging of ranges... Please explain.
Response: the blood vessel diameter to define it as a small blood vessel was presented in the introduction section (lines 109-203). Currently there are different diameters considered to define a small blood vessel from <3.0 to <2.25 mm; we used < 2.75 which is the last available information when we underwent this study and have added the reference that we used to define this diameter as the maximum lumen size to be considered a small blood vessel into the methodology section.
Is there a reason to specify the proportion of masculine gender? This was also evident in the analysis. Why not simply identify the proportion of men and women? Unclear to me why masculine gender is specified only.
Response: this was a translation error; we corrected the information to men in the abstract and in the results section.
Results
Table 2. Perhaps as a bottom footnote on the table you can identify the items that are abbreviated. DT2, AMI STU, AMI STE.... for the reader it is best to identify all abbreviations.
Response: A bottom footnote with abbreviations has been added to both table 1 and 2.
figure 1 line 172- Typo (I make this one all of the time).. "form" vs from
Response: Thank you, this has been corrected.
Figure 2 and subsequent discussion
r=.12 is a very low correlation despite statistical significance. This makes me wonder if PCI and IVUS are actually measuring similar things, as there is virtually no relationship between the measurements. Could this be the methods of taking the measures?
I think this is a section to provide more discussion. You seem to focus on the statistical relevance (<.05) rather than the very poor correlation of two methods that are supposedly measuring the same thing. Would one expect such a low correlation?
Response: This difference in measurements has been addressed by many. We added this information into the introduction section and rephrased the first paragraph of the discussion section, we also added more information to subsequent paragraphs to better address this topic.
typo- the last two lines of the manuscript (lines 318,319) I believe are not meant to be there.
Response: Thank you, the information has been removed.
Check for minor grammar edits
Response: The manuscript has been checked for grammar errors throughout.
Reviewer 3 Report
Comments and Suggestions for Authors
The manuscript is of potential interest for cardiology readers, however it requires major English editing. Several sentences are unclear and difficult to understand. Please revise the whole text for English and eventually resubmit the manuscript. In its present form, the study lacks of interest, being the text difficult to follow.
Comments on the Quality of English LanguageExtensive editing of English language required
Author Response
The manuscript is of potential interest for cardiology readers, however it requires major English editing. Several sentences are unclear and difficult to understand. Please revise the whole text for English and eventually resubmit the manuscript. In its present form, the study lacks of interest, being the text difficult to follow.
Response: The entire manuscript went through extensive English editing; it was checked for spelling and grammar errors throughout. Additionally, some sections like Abstract and introduction, went through extensive changes.
Round 2
Reviewer 1 Report
Comments and Suggestions for Authors
Thank you for making corrections to your manuscript. As for the Bland-Altman plot, I think, based on your explanation that this plot may not be suitable for your data. Anyway, I will provide an explanation for some main steps for this analysis for some of your future researches: For drawing this plot you should ensure your data does not have any missing values for the two measurements. First step is to remove rows with missing values and then calculate the mean of the two measurements and the differences between them. Then you should calculate the mean difference and the limits of agreement (mean difference ± 1.96 multiplied with SD of the differences). After those steps you can create the Bland-Altman plot with lines for the mean difference and the limits of agreement.
Author Response
Response: We would like to thank you for your time and thoughtful considerations. With your explanation, we were able to provide Bland-Altman plot, based on this, our paper now presents the information in a clearer form, we really appreciate this. Changes were made to the abstract, methods and results section. We decided to keep the correlation, since it provides a different type of information.
Reviewer 2 Report
Comments and Suggestions for Authors
Thank you for your responses to my comments and questions I have only 2 primary questions remaining:
Reporting of Male data. I am confused by how the results are tabulated. You are indicating that 70.8% of the subjects were male, so I am assuming that the data include also 29.2% female data? Is this true? Or to men represent 100% of the data you are reporting? If data from women are also in this set, then they should be somehow represented. if they are not in the reported data set, then all of the data are men. Please clarify the data set you are working with in respect to sex, and whether you are only examining one sex.
(note table 1. typo? sample size shows N=48n(%) unclear)
Discussion
A secondary purpose of the study was to examine the relationship (r) between the two methods of vascular diameter measurement. Therefore, one would expect more of a discussion on the low r seen between the variable. The first few sentences of the discussion appear to address this, but in a very cursory manner, with no real explanation for a reader to understand why the r is low. Since conclusions suggest IVUS as a superior method, one would hope for more explanation as to why the two methods do not correlate well and why IVUS is superior. It seems like an argument between gold standards, with the burden on IVUS.
Comments on the Quality of English LanguageCheck for minor typos
Author Response
Response to comment 1: We have more clearly indicated that 70.8% of the participants were men and 29.2% were women.
Response to comment 2: We made a change to the table. The information was informative with respect to the form the information was provided, instead of placing it in the table, we removed it and stated this as a footnote.
Response to comment 3: We’ve edited the first paragraph and moved some paragraphs that were written in the discussion to try to address these discrepancies, now located after this first paragraph.
Reviewer 3 Report
Comments and Suggestions for Authors
Authors have addressed my comments, and the manuscript has been improved.
Author Response
Thank you for your time in reviewing our manuscript.
Round 3
Reviewer 2 Report
Comments and Suggestions for Authors
Please add information on demographics for women to table 1. Currently it still only shows men. You are leaving out part of the same demographics in table 1.
Author Response
The table has been updated to add the number (%) of women. Although this information is typically inferred by readers.